# Learning Efficient and Robust Ordinary Differential Equations via Diffeomorphisms

## Abstract

Advances in differentiable numerical integrators have enabled the use of gradient descent techniques to learn ordinary differential equations (ODEs), where a flexible function approximator (often a neural network) is used to estimate the system dynamics, given as a time derivative. However, these integrators can be unsatisfactorily slow and unstable when learning systems of ODEs from long sequences. We propose to learn an ODE of interest from data by viewing its dynamics as a vector field related to another *base* vector field via a diffeomorphism (i.e., a differentiable bijection). By learning both the diffeomorphism and the dynamics of the base ODE, we provide an avenue to offload some of the complexity in modelling the dynamics directly on to learning the diffeomorphism. Consequently, by restricting the base ODE to be amenable to integration, we can speed up and improve the robustness of integrating trajectories from the learned system. We demonstrate the efficacy of our method in training and evaluating benchmark ODE systems, as well as within continuous-depth neural networks models. We show that our approach attains speed-ups of up to two orders of magnitude when integrating learned ODEs.

## 1 Introduction

The problem of fitting an ordinary differential equation (ODE) to observed data is ubiquitous throughout many disciplines of the natural sciences and engineering (Perko, 1991). Although traditional approaches have focused on fixed-form system dynamics and inferring their parameters, here we consider the more general problem of learning ODEs, when their dynamics are completely unknown. This problem arises, for example, in robot learning where ODEs are often used to parameterize learned motion (Sindhwani et al., 2018; Singh et al., 2020). In deep learning, this problem appears within the context of *Neural ODEs* (Chen et al., 2018), a family of continuous-depth models where the evolution of hidden states is an ODE. Recent developments in learning ODEs have allowed the use of differentiable adaptive step-size numerical integrators to train neural network dynamics via the adjoint method (Chen et al., 2018).

The learned ODE system allows us to integrate continuous trajectories at different initial conditions. To roll out long and complicated trajectories with a numerical integrator, the neural network dynamics model is queried sequentially at each step. This can be unsatisfactorily slow for time critical applications, such as those in robot control, and is known to suffer from numerical instabilities (Gholami et al., 2019; Choromanski et al., 2020). With these challenges in mind, we propose an alternative approach to learning ODEs: we view the desired target ODE dynamics as a vector field that is a "morphed" version of an alternative base vector field via a *diffeomorphism*, i.e., a bijective mapping where both the mapping and its inverse are differentiable. Thus, instead of directly modelling the time derivative of the desired ODE with a neural network, we use an invertible neural network to learn the diffeomorphism that *relates* the target ODE

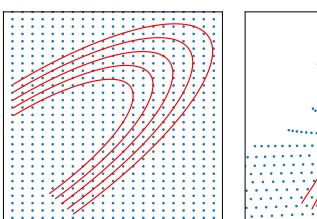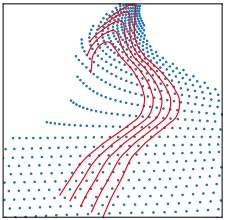

Figure 1: Related vector fields can be thought of as a vector field "morphed" into another. (Left) Five integral curves (red) of a vector field of a Linear ODE, overlaid on grid points (blue); (Right) Corresponding "morphed" integrals of the related vector field and grid.

to a base ODE. Crucially, by restricting the base ODE to be less complex and more amenable to integration, we can obtain a solution to the (more complex) target ODE by integrating the simpler base ODE and passing its solution through the bijection. Figure 1 shows an example of related vector fields: the simple integral curves in the left figure are passed through a diffeomorphism to give corresponding complicated curves in the right. Unlike rolling out a trajectory, evaluating the diffeomorphism needs not to be sequential, allowing for efficient GPU computation.

We investigate restricting the flexibility of the base ODE to improve integration efficiency, and offloading more of the representation burden to the diffeomorphism. Specifically, we assume the dynamics of the base ODE to be: (1) linear or (2) modelled by a neural network. Restricting the base ODE to be linear allows the computation of closed-forms solutions, providing major speed benefits. In this set-up we can achieve significant speed-ups of up to two orders of magnitude when employing GPUs, as compared against differentiable integrators with standard settings. We can also restrict the learned ODE to be provably asymptotically stable by adding simple constraints to the linear base ODE. Alternatively, when additional flexibility is required, we remove the restrictive assumption of a linear base ODE and model the dynamics using a neural network. We show that in this setting we can improve the performance of learning challenging ODEs compared to existing differentiable integrators, even when we use a simpler neural network for the base ODE.

In summary, our main contributions are:

1. a novel paradigm to learn ODEs from data: invertible neural networks are trained to "morph" the target ODE to an alternative *related* base ODE, which can be more tractably integrated;

2. analysis of the base as (i) a linear ODE and (ii) a non-linear ODE with neural network dynamics. In the linear case, we demonstrate how by restricting the flexibility of the base ODE, we can obtain closed-form integrals, providing significant speed-ups to integration. In the non-linear case, we demonstrate that by learning ODEs as related vector fields, we can flexibly learn challenging ODEs with simpler networks;

3. a principled method to enforce asymptotic stability of learned ODEs, by adding restrictions to the base ODE.

Proofs and additional details can be found in the appendices.

## 2 RELATED WORK

**Learning of ODEs and Neural ODEs:** Dynamical systems governed by ODEs can be found throughout many disciplines of science and engineering. Earlier work on approximating free-form dynamics of ODEs include gradient matching (Ramsay et al., 2006) and using Gaussian processes (Heinonen et al., 2018). Most recent work on this problem model the unknown dynamics with a neural network and leverage differentiable numerical integrators, which use the adjoint method (Pontryagin et al., 1962) to train in a memory-usage tractable manner (see, e.g., Chen et al., 2018).

A particular usage of ODE learning is within Neural ODEs, which are neural network models that model the hidden state as continuous ODEs rather than discrete layers (Chen et al., 2018; Massaroli et al., 2020). Other continuous neural network models which incorporate an ODE, such as latent ODEs (Rubanova et al., 2019) have found application in time-series tasks. Subsequent strategies have been introduced to improve the training of these models, including augmenting the ODE state-space (Dupont et al., 2019), hyper-network extensions (Choromanski et al., 2020), regularisation techniques (Finlay et al., 2020; Pal et al., 2021), and investigating integrator step-sizes (Ott et al., 2021). At the core of all neural ODE models is the differentiable integrator used to learn the underlying ODE. Our proposed approach improves the learning of the underlying ODE, and is compatible with models that incorporate learnable ODEs. We note that the term "neural ODE" has typically been used in the literature to refer to neural networks that incorporate ODEs, including the original work in Chen et al. (2018). However, "neural ODE" has occasionally been used to refer to an ODE with dynamics parameterized by a neural network (Norcliffe et al., 2021). To disambiguate, throughout our paper, we refrained from referring to the latter model as "neural ODEs", but rather as "ODEs with dynamics parameterized by a neural network".

**Invertible neural networks and Normalizing Flows:** Invertible neural networks (INNs) are a class of function approximators that learn bijections where the forward and inverse mapping and their

Jacobians can be efficiently computed (Ardizzone et al., 2019). INNs are typically constructed by invertible building blocks, such as those introduced in Kingma et al. (2016); Dinh et al. (2017); Durkan et al. (2019). Advances in INNs are largely motivated by normalizing flows (Rezende & Mohamed, 2015; Papamakarios et al., 2021), an approach to construct a flexible probability distribution by finding a differentiable bijection between the target distribution and a base distribution. Our approach is similar in spirit to normalizing flows, as we analogously aim to learn a diffeomorphism that relates the vector fields of the target ODE and some base ODE. However, unlike normalizing flows, we do not require the burdensome computation of Jacobian determinants (Karami et al., 2019) to obtain trajectories. A separate line of work, broadly characterized as *continuous normalizing flows*, use ODEs to build invertible approximators (Grathwohl et al., 2019; Chen et al., 2018). Our work proposes the opposite where invertible approximators are used to learn ODEs.

## 3 PRELIMINARIES

In this section we introduce the problem of learning ODE dynamics with neural networks. We then describe the notions of tangent spaces and pushforwards, which will be used in section 4 to develop our method.

### 3.1 LEARNING ODES WITH NEURAL NETWORKS

Consider a dynamical system given by ordinary differential equations of the form:

$$\mathbf{y}'(t) = f(\mathbf{y}(t), t), \qquad\qquad \mathbf{y}(0) = \mathbf{y}_0, \qquad\qquad (1)$$

where $t$ is time, $\mathbf{y}(t)$ are the states at time $t$, and $f$ provides the dynamics. Unlike traditional approaches where $f$ is assumed known with only a few parameters to estimate from data, here we consider the more general problem where the dynamics are completely unknown. Thus, we can use a flexible mapping $f_\omega$ as given by a neural network with parameters $\omega$. Henceforth, we will drop the explicit dependence on time, and consider the autonomous ODEs given by $\mathbf{y}'(t) = f(\mathbf{y}(t))$. Non-autonomous ODEs, which explicitly depend on time, can be equivalently expressed as autonomous ODEs by adding a dimension to the states $\mathbf{y}$ (Davis et al., 2020). For an initial condition $\mathbf{y}_{t_0}$ at start $t_0$, and some end time $t_e$, a solution of the ODE can be evaluated by a numerical integrator (ODESolve), such as Runge-Kutta methods (Butcher, 1987):

$$\mathbf{y}(t_e) = \mathbf{y}_{t_0} + \int_{t_0}^{t_e} f_\omega(\mathbf{y}(t)) \mathrm{d}t = \mathrm{ODESolve}(f_\omega, \mathbf{y}_{t_0}, t_e - t_0). \qquad\qquad (2)$$

The learning problem involves estimating, with $f_\omega$, the dynamics of the ODE, provided $n_t$ observations $\mathbf{y}_{t_1}^{obs} \ldots \mathbf{y}_{t_{n_t}}^{obs}$ at specified times. We can learn the ODE by optimising the parameters $\omega$ to minimize a loss between the observations at the given times and the integrated ODE, $\ell(\omega) = \mathrm{Loss}(\{\mathbf{y}_{t_i}^{obs}\}_{i=1}^{n_t}, \{\mathbf{y}(t_i)\}_{i=1}^{n_t})$, where $\{\mathbf{y}(t_i)\}_{i=1}^{n_t}$ are obtained by solving eq. (2). Advances in the neural ODE literature have introduced differentiable numerical integrators, which allow gradient-based techniques to be used to optimize $l(\omega)$. Furthermore, by using the adjoint sensitivity method as outlined in Chen et al. (2018), the gradients of adaptive integrators can be obtained in a memory tractable manner, without differentiating through the integrator operations.

Nevertheless, the flexibility of neural network dynamics for ODE learning comes at the expense of a high-computational cost and potential numerical instabilities, especially when considering long trajectories. We will develop in section 4 an alternative method that transforms this problem into that of learning a simpler ODE along with a diffeomorphism, by treating the dynamics of the original (target) ODE and simpler (base) ODE as related vector fields.

### 3.2 TANGENT SPACES AND PUSHFORWARDS

As mentioned above, we shall be analysing the system dynamics of ODEs as vector fields. Here we briefly introduce the differential geometry notions of tangent spaces and pushforwards, which will be used to define *related vector fields*, a core concept underpinning our methodology.

**Tangent Spaces:** A manifold is a space that locally resembles Euclidean space. Throughout this paper, all manifolds will be assumed to be differentiable, with defined *tangent spaces*. For an

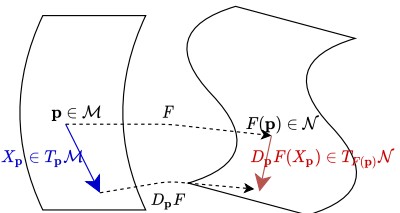 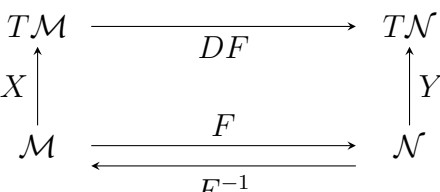

(a) If $F$ maps points $\mathbf{p} \in \mathcal{M}$ to $F(\mathbf{p}) \in \mathcal{N}$, a single tangent vector at $\mathbf{p}$, $X_{\mathbf{p}} \in T_{\mathbf{p}}\mathcal{M}$, can be mapped to $T_{F(\mathbf{p})}\mathcal{N}$. However, an entire vector field $X$ on $\mathcal{M}$ *cannot* in general be mapped to a valid vector field on $\mathcal{N}$. The pushforward by a diffeomorphism is a special case where a valid vector field can be obtained.

(b) If vector fields $X$ and $Y$ on manifolds $\mathcal{M}$ and $\mathcal{N}$ respectively are *related* by diffeomorphism $F$, then they are related via the pushforward of $F$. If $Y$ is unknown, we have another path to evaluate $Y$ by $\mathcal{N} \xrightarrow{F^{-1}} \mathcal{M} \xrightarrow{X} T\mathcal{M} \xrightarrow{DF} T\mathcal{N}$.

Figure 2: Vector fields can be related by a diffeomorphisms

$n$-dimensional manifold $\mathcal{M}$, at a point $\mathbf{p} \in \mathcal{M}$, the tangent space $T_{\mathbf{p}}\mathcal{M}$ is an $n$-dimensional real vector space, where each element passes $\mathbf{p}$ tangentially and is referred to as a *tangent vector*. The tangent space provides a higher-dimensional analogue of a tangent plane at a point on a surface. The collection of tangent spaces for all points on $\mathcal{M}$ is known as the *tangent bundle* denoted by $T\mathcal{M}$.

**Pushfoward:** For a mapping $F : \mathcal{M} \to \mathcal{N}$ between two manifolds, $\mathcal{M}$ and $\mathcal{N}$, the *pushforward* by $F$ is a linear mapping between the tangent spaces of the manifolds, $D_{\mathbf{p}}F : T_{\mathbf{p}}\mathcal{M} \to T_{F(\mathbf{p})}\mathcal{N}$. Tangent vectors at $\mathbf{p}$ in the domain $\mathcal{M}$ can be mapped to tangent vectors at the corresponding point $F(\mathbf{p})$ in the codomain $\mathcal{N}$ via the pushforward. This is computed by the matrix product of the Jacobian of $F$ at $\mathbf{p}$, $J_F(\mathbf{p})$, and a tangent vector at $\mathbf{p}$.

## 4 METHODOLOGY

We study the dynamics of ODEs, $f$, as vector fields, and solutions as their integral curves. We model the desired ODE dynamics as a *target* vector field that is *related* to another *base* vector field. First, we introduce the concept of related vector fields, outline how they can be learned, and elaborate on the benefits of learning them. Then, we describe possible choices of base vector field models.

### 4.1 RELATED VECTOR FIELDS FOR ODE LEARNING

A vector field $X$ defined on manifold $\mathcal{M}$ is a function that assigns a tangent vector $X_{\mathbf{p}} \in T_{\mathbf{p}}\mathcal{M}$ to each point $\mathbf{p} \in \mathcal{M}$. Intuitively, our aim is to construct a mapping $F$ which shapes the manifold where a *base* vector field $X$ is defined, such that the pushforward of $X$ by $F$ extrinsically appears "morphed" to match the data.

***What are the requirements of these mappings, for the pushforward of vector fields to be valid?***

Provided a mapping between manifolds $F : \mathcal{M} \to \mathcal{N}$, we can push a single vector, $X_{\mathbf{p}} \in T_{\mathbf{p}}\mathcal{M}$, to the tangent space of $\mathcal{N}$ at $F(\mathbf{p})$, $T_{F(\mathbf{p})}\mathcal{N}$, via the *pushforward*, $D_{\mathbf{p}}F(X_{\mathbf{p}})$. Figure 2a sketches out an example of the pushforward of a tangent vector between tangent spaces. However, this notion does not extend in general to vector fields. If $F$ is injective and non-surjective, the pushforward of $X$ outside the image of $F$ is not defined. If $F$ is surjective and non-injective, there may be multiple differing pushforwards for a point. In special cases when the pushforward by $F$ defines a valid vector field on the codomain $\mathcal{N}$, the vector field and its pushforward are known to be $F$-related. Latent ODEs, as described in (Rubanova et al., 2019), which use an auto-encoder to map input data to a latent space, where an ODE is learned is similar in spirit to our method. Latent ODEs, however, do not define a valid vector field in the input space.

**Definition 4.1** (Related vector fields)**.** *Let $F : \mathcal{M} \to \mathcal{N}$ be a smooth mapping of manifolds. A vector field $X$ on $\mathcal{M}$ and a vector field $Y$ on $\mathcal{N}$ are related by $F$, or $F$-related, if for all $\mathbf{p} \in \mathcal{M}$,*

$$D_{\mathbf{p}}F(X_{\mathbf{p}}) = Y_{F(\mathbf{p})}. \tag{3}$$

**Related vector fields arise in particular when $F$ is a diffeomorphism,** i.e. a bijective mapping, where both the mapping itself and its inverse are differentiable.

**Proposition 4.1** (Proposition 8.19 in Lee (2012)). *Suppose $F : \mathcal{M} \to \mathcal{N}$ is a diffeomorphism between smooth manifolds $\mathcal{M}, \mathcal{N}$. For every vector field $X$ on $\mathcal{M}$, there is a unique vector field $Y$ on $\mathcal{N}$ that is $F$-related to $X$.*

By considering the $F$-related properties of vector fields, we have a pathway to define unknown vector fields using the pushforward of $F$, as shown in fig. 2b. If vector field $X$ on $\mathcal{M}$ is $F$-related to some vector field $Y$ on $\mathcal{N}$, instead of directly evaluating the vector field $Y$, we can instead obtain tangent values for any $\mathbf{q} \in \mathcal{N}$, via $\mathcal{N} \xrightarrow{F^{-1}} \mathcal{M} \xrightarrow{X} T\mathcal{M} \xrightarrow{DF} T\mathcal{N}$. Therefore, the vector attached for each $\mathbf{q} \in \mathcal{N}$ is, $Y_{\mathbf{q}} = D_{F^{-1}(\mathbf{q})}F(X_{F^{-1}(\mathbf{q})}) = J_F(F^{-1}(\mathbf{q}))X_{F^{-1}(\mathbf{q})}$, where $J_F$ is the Jacobian of $F$.

## 4.2 DIFFEOMORPHISM LEARNING VIA INVERTIBLE NEURAL NETWORKS

The machinery to learn invertible mappings has seen extensive development with the progress of normalizing flows for estimating probability distributions. Invertible neural networks (INNs, Ardizzone et al., 2019) are function approximators which learn differentiable bijections. INNs can be trained on a forward mapping, and get the inverse with no additional work, by the definition of their architecture. In this paper, we use INNs of the type described in Dinh et al. (2017). The basic unit is a reversible block, where inputs are split into two halves, $\mathbf{u}_1$ and $\mathbf{u}_2$. The outputs $\mathbf{v}_1$ and $\mathbf{v}_2$ are:

$$\mathbf{v}_1 = \mathbf{u}_1 \odot \exp(s_2(\mathbf{u}_2)) + t_2(\mathbf{u}_2), \qquad \mathbf{v}_2 = \mathbf{u}_2 \odot \exp(s_1(\mathbf{v}_1)) + t_1(\mathbf{v}_1), \qquad (4)$$

where $\odot$ indicates element-wise multiplication, and $t_1$, $t_2$ and $s_1$, $s_2$ are functions modelled by fully-connected neural networks with non-linear activations. These expressions are clearly invertible:

$$\mathbf{u}_1 = (\mathbf{v}_1 - t_2(\mathbf{u}_2)) \odot \exp(-s_2(\mathbf{u}_2)), \qquad \mathbf{u}_2 = (\mathbf{v}_2 - t_1(\mathbf{v}_1)) \odot \exp(-s_1(\mathbf{v}_1)). \qquad (5)$$

Note that the functions $t_1$, $t_2$ and $s_1$, $s_2$ themselves are not required to be invertible.

## 4.3 ODE SOLUTIONS AS INTEGRAL CURVES OF RELATED VECTOR FIELDS

***Why would it be beneficial to construct a desired vector field $Y$ indirectly, by way of a related $X$?***

We shall answer this by considering integral curves on $Y$, which represent solutions to the ODE associated with $Y$. An integral curve of $Y$ on $\mathcal{N}$ is a differentiable curve $\mathbf{y} : \mathbb{R} \to \mathcal{N}$, whose velocity at each point is equal to the value of $Y$ at that point, i.e. $\mathbf{y}'(t) = Y_{\mathbf{y}(t)} \in T_{\mathbf{y}(t)}\mathcal{N}$, for all $t \in \mathbb{R}$. The integral curves of $F$-related vector fields are also linked by $F$: integral curves on one vector field are *uniquely* mapped to the other via a single pass through $F$, and the Jacobian $J_F$ is not required.

**Proposition 4.2** (Proposition 9.6 in Lee (2012)). *Suppose $X$ and $Y$ are vector fields on manifolds $\mathcal{M}$ and $\mathcal{N}$ respectively. $X$ and $Y$ are related by mapping $F : \mathcal{M} \to \mathcal{N}$ if and only if for each integral curve $\mathbf{x} : \mathbb{R} \to \mathcal{M}$, $\mathbf{y} = F(\mathbf{x})$ is an integral curve of $Y$.*

In the ODE learning problem outlined in section 3.1, we denote $Y$ to be the vector field associated with the target ODE. During both training and inference, we need to obtain integral curves $\mathbf{y}$ of $Y$ either by numerical integration, or by $\mathbf{y} = F(\mathbf{x})$, where $\mathbf{x}$ denotes the corresponding integral curve of $X$, related to $Y$ via the diffeomorphism $F$. Critically, the Jacobian of $F$ does not need to be evaluated when we are working with the integral curves.

If integral curves of $X$ can be found in a more efficient, or less error-prone manner, than by numerically integrating curves of $Y$, we can leverage the relationship $\mathbf{y} = F(\mathbf{x})$ for ODE learning. This can be done by an INN, $F_\theta$, with parameters $\theta$. We denote the base ODE as $\mathbf{x}'(t) = g_\varphi(\mathbf{x}(t))$, where $\varphi$ are parameters. We can then use the target ODE within some learning problem, minimising a loss over target ODE trajectories and observations:

$$\ell(\theta, \varphi) = \text{Loss}\Big(\big\{\mathbf{y}_{t_i}^{obs}\big\}_{i=1}^{n_t}, \big\{F_\theta\big(F_\theta^{-1}(\mathbf{y}_0) + \int_0^{t_i} g_\varphi(\mathbf{x}(t))\mathrm{d}t\big)\big\}_{i=1}^{n_t}\Big), \qquad (6)$$

where there are $n_t$ data time points, $\mathbf{y}_0$ is an initial condition for the system, and $\mathbf{y}_{t_i}^{obs}$ are observed data points at times $t_i$. After training, the dynamics of the target ODE is given by $\mathbf{y}'(t) = J_{F_\theta}\big(F_\theta^{-1}(\mathbf{y}(t))\big)g_\varphi\big(F_\theta^{-1}(\mathbf{y}(t))\big)$ and the ODE solutions (integrals) are obtained with:

$$\mathbf{y}(t) = F_\theta\big(F_\theta^{-1}(\mathbf{y}_0) + \int_0^t g_\varphi(\mathbf{x}(t))\mathrm{d}t\big). \qquad (7)$$

In practice, we are often required to evaluate an entire trajectory, i.e., $\mathbf{y}(t)$ at multiple times $t_1, \ldots, t_{end}$ with one initial $\mathbf{y}_0$, as outlined in Algorithm 1. This allows us to batch up the pass through $F_\theta$, which makes this highly efficient when executed on a GPU. The benefits of our method are apparent when it is advantageous to integrate the base ODE and then pass the solution through the diffeomorphism, $F_\theta$, rather than numerically integrate the target ODE.

Next, we investigate restricting the flexibility of the base ODE, so that it is more amenable to integration, offloading the complexity of learning to the diffeomorphism. We consider two choices of base ODEs: (1) Linear ODE; (2) ODE with neural network dynamics.

---

**Algorithm 1:** Efficient integration of learned ODEs

**Input** : $F_\theta, g_\varphi, \mathbf{y}_0, t_1, \ldots, t_{end}$
**Output**: $\mathbf{y}(t_1), \ldots, \mathbf{y}(t_{end})$

1 $\mathbf{x}_0 \leftarrow F_\theta^{-1}(\mathbf{y}_0)$
2 $\mathbf{x}(t_i) \leftarrow \mathbf{x}_0 + \int_0^{t_i} g_\varphi(\mathbf{x}(t))\mathrm{d}t$, for $i = 1, \ldots, end$ ;
    // The integral is easier to solve.
3 $\mathbf{y}(t_1), \ldots, \mathbf{y}(t_{end}) \leftarrow F_\theta(\mathbf{x}(t_1), \ldots, \mathbf{x}(t_{end}))$ ;
    // Batched pass through INN can be efficiently computed on GPUs.

---

### 4.4 LINEAR ODE AS BASE: FAST INTEGRATION AND STRAIGHTFORWARD ASYMPTOTIC STABILITY

We can speed-up integration significantly by modelling the base as a Linear ODE, of the form $\mathbf{x}'(t) = A\mathbf{x}(t)$, where $\mathbf{x}(t) \in \mathbb{R}^n$ are $n$-dimensional variables, and $A \in \mathbb{R}^{n \times n}$. Linear ODEs can be solved very efficiently as they admit closed-form solutions. Provided an initial solution $\mathbf{x}_0$, the solution of $\mathbf{x}(t)$ and that of the target ODE $\mathbf{y}(t)$ are: $\mathbf{x}(t) = \sum_{k=1}^n (\mathbf{l}_k \cdot \mathbf{x}_0)\mathbf{r}_k \exp(\lambda_k t)$, and $\mathbf{y}(t) = F_\theta(\mathbf{x}(t))$,

where $\mathbf{l}_k$, $\mathbf{r}_k$ and $\lambda_k$ are the corresponding left, right eigenvectors and eigenvalues of matrix $A$, respectively. We learn the eigenvalues and eigenvectors of matrix $A$ jointly with diffeomorphism $F_\theta$.

Linear ODEs are also interesting because their long-term behavior, which is determined by their eigenvalues, is easy to analyse. We shall see how this property allows us to craft the long-term behavior of the target ODE. In particular, in many applications, consideration is given to the asymptotic properties of ODEs, namely what happens to the solutions after a long period of time. Will they converge to equilibrium points, periodic orbits, or diverge and fly off? Our method provides a straightforward way to restrict the learned ODE to be asymptotically stable. In robot motion generation problems, such as that in Sindhwani et al. (2018), we aim to learn an asymptotically stable ODE. We begin by defining equilibrium points and asymptotic stability of first order ODEs.

**Definition 4.2** (Equilibrium point). *An equilibrium point $\mathbf{y}^*$ of an ODE $\mathbf{y}'(t) = f(\mathbf{y}(t))$, is a point where $f(\mathbf{y}^*) = 0$.*

**Definition 4.3** (Asymptotic stability). *An ODE $\mathbf{y}'(t) = f(\mathbf{y}(t), t)$ is asymptotically stable if for every solution $\mathbf{y}(t)$, there exists a $\delta > 0$, such that whenever $||\mathbf{y}(t_0) - \mathbf{y}^*|| < \delta$, then $\mathbf{y}(t) \to \mathbf{y}^*$ as $t \to \infty$, where $\mathbf{y}^*$ is some equilibrium point.*

Intuitively, asymptotically stable systems of ODEs will always settle at some equilibrium points after a long period of time. In the context of vector fields related by a diffeomorphism, the asymptotic stability properties of the ODEs are shared.

**Theorem 4.1.** *Suppose two ODEs $\mathbf{x}'(t) = g(\mathbf{x}(t))$, $\mathbf{y}'(t) = f(\mathbf{y}(t))$ are related via $\mathbf{y}(t) = F(\mathbf{x}(t))$, where $F$ is a diffeomorphism. If the former ODE is asymptotically stable with $n_e$ equilibrium points $\mathbf{x}_1^*, \ldots, \mathbf{x}_{n_e}^*$, then the latter is also asymptotically stable, with equilibrium points $F(\mathbf{x}_1^*), \ldots, F(\mathbf{x}_{n_e}^*)$.*

Therefore, if we can restrict the base ODE to be asymptotically stable, then the target ODE learned by our method is also asymptotically stable. When the base is an $n$ dimensional linear ODE, we can restrict it to be asymptotically stable by directly learning the eigenvalues, $\lambda_i$ for $i = 1, \ldots, n$, and constraining them to be negative. This can be done by setting $\lambda_i = -(s_{\lambda_i})^2 - \varepsilon$, where $\varepsilon$ is a small positive constant, and learning $s_{\lambda_i}$ instead of learning the eigenvalues.

### 4.5 NEURAL NETWORK ODE AS BASE: IMPROVED ROBUSTNESS FOR 'DIFFICULT' ODES

Using linear systems as base ODEs provides a dramatic increase in speed at the cost of flexibility. We observe that the computation overhead of a single backward pass $F_\theta^{-1}$ and a batched single

forward pass $F_\theta$ is minimal when compared with numerical integration which requires sequential querying. When the ODE is difficult to learn, we can also parameterize the dynamics of the base ODE using a neural network. This is particularly appealing for ODEs that are considered stiff, with rapid varying of the solution in time in one dimension, while the other dimensions remain largely unchanged. Existing differentiable integrators often struggle to directly learn these ODEs. Directly learning the target ODE will require exceedingly small step-sizes (Hairer et al., 1993). Although by simply scaling the data before training, we can lessen the stiffness of the ODE to learn, the relatively rapid changes isolated in a single dimension can still result in the ODE being hard to learn. To tackle this, we learn a neural network dynamics of the base ODE jointly with the diffeomorphism. The burden of accurately representing the stiff dynamics is shared by $F_\theta$, providing additional flexibility. The diffeomorphism can learn to relate the target ODE to a base that is amenable to integration. This allows us to more accurately learn challenging ODEs, with a much smaller neural network model of the base ODE, than that used to directly learn the target ODE. As passes of the INN are inexpensive, we gain a speed-up during integration.

## 5 EXPERIMENTAL RESULTS

We empirically evaluate the ability of our method to speed up the integration of learned ODEs, along with the robustness of integration when learning difficult ODE systems. Throughout this section, we compare the error and integration times of our method against a variety of solvers. We include fixed step-size solvers: Euler's, midpoint, and Runge-Kutta 45 (RK4), and the adaptive step-size solvers Dormand–Prince 5 (DOPRI5) and Dormand–Prince 8 (DOPRI8). Unless specified otherwise, for fixed step-size solvers, we set the step-size equal to the smallest time increment of outputs. For adaptive step-size solvers, we set absolute and relative tolerances to $10^{-5}$. We augment the ODE states for all the systems in accordance to Dupont et al. (2019) in all of the ODEs trained during the experiments, except when recreating latent ODE results from Rubanova et al. (2019), where we use the original implementation provided by the authors. The differentiable solvers are implemented in the *torchdiffeq* library. Details on experiments available in the appendices.

### 5.1 INTEGRATION SPEED-UPS BY LEARNING WITH A LINEAR ODE BASE

We test our hypothesis that the availability of a closed-form expression for the integral, when using a linear base ODE, can provide significant integration speed-ups. We evaluate on learning synthetic ODE systems, real-world robot demonstrations, and within continuous deep learning models. Here, we report test error/accuracy and integration times. Training times can be found in the appendices.

**Learning 3D Lotka-Volterra:** We train and evaluate models on data from the 3D Lokta-Volterra system, which models the dynamics of predator-prey populations. The data is corrupted by white noise with standard deviation of $0.05$. We train our model using a linear base ODE, and assess the capability of our model in interpolating the data points at 10x the data resolution, and generalizing to 16 hidden test initial conditions to integrate trajectories, also at 10x the data resolution. We report the integration time for generalization. Figure 3 shows interpolation results and newly generated trajectories, where we see that our model is able to capture the dynamics of the system. Furthermore, table 1 provides a quantitative evaluation, where we see that our method is significantly faster

| | 3D Lotka-Volterra | | | Imitation S | | Imitation cube pick | | Imitation C | |
|---|---|---|---|---|---|---|---|---|---|
| | MSE (I) | MSE (G) | Time (ms) | MSE (G) | Time (ms) | MSE (G) | Time (ms) | MSE (G) | Time (ms) |
| Ours (Lin) | 0.14± 0.1 | 1.5± 0.1 | 9.3± 0.4 | 6.1±1.2 | 6.6± 0.2 | 18.6± 6.2 | 7.1 ± 1.6 | 8.1± 1.6 | 7.5± 0.8 |
| Euler | 4.5± 0.3 | 4.6± 0.1 | 385.6± 14.4 | 10.3±2.9 | 724.7± 8.3 | 14.9± 1.4 | 728.4± 9.5 | 7.3± 2.0 | 753.9± 1.4 |
| Midpoint | 0.38± 0.05 | 5.51± 0.1 | 670.4± 31.3 | 10.9±3.3 | 581.6± 13.3 | 12.9± 1.3 | 1267.2± 13.6 | 6.9± 2.2 | 1305.4± 14.7 |
| RK4 | 0.35± 0.005 | 5.6± 0.2 | 1316.1± 30.8 | 10.3±3.0 | 2501.7± 18.9 | 15.9± 0.9 | 2522.8± 23.1 | 7.6± 2.7 | 1292.3± 22.0 |
| DOPRI5 | 0.93± 0.05 | 5.19± 0.5 | 264.7± 17.0 | 10.8±2.8 | 1277.7± 14.3 | 14.9± 0.9 | 504.0± 12.3 | 7.1± 1.9 | 623.4± 15.6 |

Table 1: The mean squared error for interpolation, MSE (I), and generalization, MSE (G), and mean execution times (in ms, ± 1 standard deviation) on the 3D Lotka-Volterra system and the time critical motion generation for our method with a linear base ODE and competing augmented ODE models, with neural network dynamics, with various numerical integrators. Our method, with a linear base ODE, provides comparable or better accuracy, with significant integration speed-ups.

than competing approaches using numerical integrators with speed-ups of more than two orders of magnitude, while achieving comparable or better accuracy.

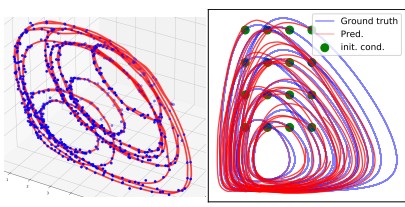

Figure 3: Learning the 3D Lotka–Volterra. (Left) Interpolating (red) data (blue); (Right) Generating trajectories (red) at hidden test initial conditions and the ground truth (blue).

**A Time Critical Application—Robot Motion Generation as Stable ODE Learning:** The ability to quickly roll-out trajectories is crucial in motion robotics settings. We consider the application of generating robot manipulator motion trajectories from provided demonstrations. In particular, Sindhwani et al. (2018) showed that modelling the motion as trajectories of a stable ODE is critical for generalizing and being robust to perturbations in initial conditions. The goal is to learn an ODE system where trajectories integrated at different starting points mimic the shown demonstrations. We use three sets of real-world data from (Khansari-Zadeh & Billard, 2011) of trajectories: drawing "S" shapes; placing a cube on a shelf; drawing out large "C" shapes. We use $70\%$ of the data for training, and test our generalization on the remaining demonstrations.

In these datasets, the motions are known to converge to equilibrium points. Hence, we constrain the learned ODE to be asymptotically stable. We report the performance and run-times of generalizing to new starting points in table 1. We see that our approach is competitive in the quality of generalized trajectories, while achieving speed-ups of more than two-orders of magnitude.

**ODE Learning for Continuous Deep Learning Models:** We evaluate our method as a component of *Latent ODEs* (Rubanova et al., 2019), a continuous deep learning model. Latent ODEs embed the time series observations as hidden states via an encoder-decoder. An ODE, with dynamics parameterized by a neural network, is fit on the hidden states which allows for irregularly sampled series. In these experiments, our method is applied with

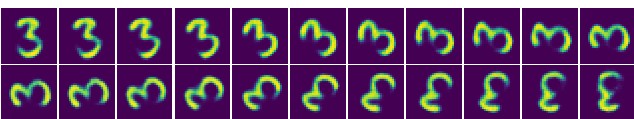

Figure 4: We model latent dynamics of rotating MNIST "3" characters, as an ODE. We learn the ODE via a diffeomorphism and a linear base ODE. Given an unseen test "3" character, we efficiently generate a image sequence of rotating "3"s (illustrated here, from upper left to lower right).

a linear ODE base to learn the dynamics governing the hidden states. We report results for the periodic curves and the human activity classification problem used in the original latent ODE paper (Rubanova et al., 2019), as well as for ECG classification and image sequence generation. The periodic curves problem requires us to reconstruct trajectories at different resolutions, with 100 and 1000 time-steps. The human activity classification dataset contains readings over the body of subjects over time, and the problem has a sequence-to-sequence setup, requiring us to predict the human activity at each time point. The ECG problem is a sequence classification problem, where the sequences are ECG signals. In the image sequence generation problem, we train on 100 sequences of rotating "3" characters, similar to that in Yildiz et al. (2019), learning a high-dimensional ODE which governs a latent representation of the image rotation over time. We test on 100 unseen "3" characters and report results at 10x training resolution. The performance and times spent on integrating the hidden state dynamics are reported in table 2. We observe that by leveraging the closed-form expression of integrals of the base ODEs, we achieve ODE integration times that are hundreds of times faster, while achieving competitive performances against compared differentiable integrators. We note that the main cost of the integral in our method is the pass through the invertible neural network. GPUs allow us to batch the pass at practically constant cost, whereas the sequential nature of integrators give linear increases in run-time. A qualitative evaluation of our method on the image sequence problem is shown in fig. 4. We observe the structure of the rotated "3" character remains consistent with the test initial image. Our approach, when restricted to a linear base ODE, is able to learn high-dimensional ODEs which can be integrated significantly faster, without compromising performance.

## 5.2 ROBUST INTEGRATION BY LEARNING WITH A NON-LINEAR NEURAL NETWORK BASE

We test our hypothesis that using a neural network base ODE allows us to learn ODEs that are difficult to integrate. We learn and evaluate models trained on the chaotic Lorenz system, and the stiff

|  | Periodic 100 | | Periodic 1000 | | Human Activity | | ECG | | Image Seq. | |
|---|---|---|---|---|---|---|---|---|---|---|
|  | MSE | Int. time | MSE | Int. time | Acc. | Int. time | Acc. | Int. time | MSE | Int. time |
| Ours (Lin) | 0.030 | 2.7± 0.6 | 0.008 | 2.8± 0.8 | 0.864 | 4.2± 1.8 | 0.966 | 7.6± 2.5 | 0.028 | 5.7 ± 1.3 |
| Euler | 0.040 | 33.7± 2.6 | 0.043 | 326.6± 9.5 | 0.815 | 67.9± 2.9 | 0.963 | 100.0± 2.8 | 0.028 | 103.3±2.5 |
| Midpoint | 0.032 | 54.5± 1.8 | 0.074 | 510.1± 15.5 | 0.865 | 114.2± 2.4 | 0.963 | 169.7± 3.8 | 0.026 | 187.9±6.8 |
| RK4 | 0.039 | 95.6± 1.6 | 0.052 | 1020.0± 60.0 | 0.857 | 221.2± 4.2 | 0.963 | 325.5± 4.8 | 0.027 | 401.2±9.1 |
| DOPRI5 | 0.045 | 83.4± 2.2 | 0.050 | 264.7± 4.6 | 0.869 | 67.9± 5.0 | 0.963 | 123.3± 2.6 | 0.025 | 194.5± 9.2 |
| DOPRI8 | 0.041 | 99.6± 2.3 | 0.049 | 282.7± 6.4 | 0.724 | 94.8± 1.6 | 0.963 | 171.6± 3.6 | 0.026 | 399±28.4 |

Table 2: The mean squared error, accuracy and mean integration times (in ms, ± 1 standard deviation) when using latent ODEs on the tasks of periodic curve reconstruction using 100 and 1000 time-steps, the classification problems of human activity and ECG, and the image sequence generation with our method using a linear base ODE and competing numerical integrators.

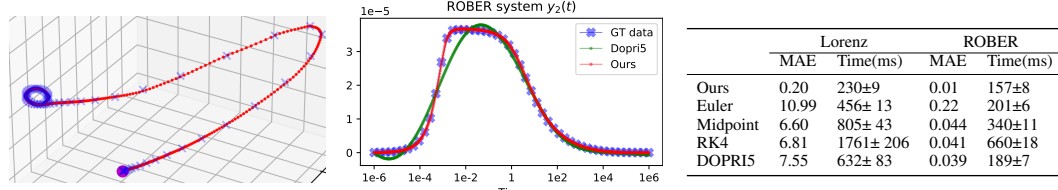

Figure 5: Results on chaotic (Lorenz) and stiff (ROBER) systems. *Left:* A trajectory from the learned Lorentz system in red, with data in blue. *Center:* The second dimension of the ROBER system, against time in logscale. Our method is much capable at capturing the sudden variation. *Right:* Quantitative evaluation of our method with a non-linear base ODE and competing integrators on augmented ODEs with neural network dynamics.

Robertson's system (ROBER, Robertson, 1966). We use a DOPRI5 adaptive step-size integrator to learn our base ODE, generate trajectories at 10x data resolution. Figure 5 (left) illustrates a generated trajectory from the Lorenz system, which closely resembles the data points (blue). We can see that both the initial large and small variations, indicated by the small dense spiral at the end, are captured. Figure 5 (center) illustrates the particular rapidly changing second dimension of the ROBER system. We see that the added flexibility of the diffeomorphism allows us to better capture the rapid variations over time, while the directly learned ODE struggles to handle the sudden increase. Following the equation rescaling described in Kim et al. (2021), before we train on our method and comparisons, we rescale our data by the maximum training value in each dimension and operate in logscale time. Figure 5 (right) provides the performance and integration times of learning with our method and baseline integrators, where we see that our method is more accurate than competing approaches while also requiring shorter integration times, due to having a simpler network modelling the dynamics.

# 6 CONCLUSIONS

We have proposed a novel approach to learning ODEs with unknown dynamics, which uses invertible neural networks to learn a diffeomorphism relating a desired target ODE to a *base* ODE that is often easier to integrate. We have investigated using a base ODE that is linear or parameterized by a neural network. By leveraging the closed form solution of linear ODEs, our method provides significant speed-ups and allows for asymptotic property constraints on the learned ODEs. Alternatively, by using a base ODE parameterized by a neural network, our approach can learn "difficult" ODEs, with simpler networks modeling their dynamics. We have validated our method by learning ODEs on synthetic and real-world data, on robotic learning problems and within continuous-depth neural network models. Future work could explore more on how to balance offloading the burden of learning to the diffeomorphism and the base ODE.

# 7 REPRODUCIBILITY STATEMENT

It is extremely important that the work published in ICLR is reproducible. To this end, we have included source code for our experiments, including code to generate the benchmark dynamical systems, as supplementary materials to the submission. Furthermore, additional details of experiments

and data, along with clear explanations of any assumptions and a complete proof of the theoretical claims are included in the appendix.

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

The following sections contain the supplementary text which gives additional results, additional details about the experiments, and proofs.

## A    ADDITIONAL RESULTS: REPRESENTATION LOAD OF THE INN

In the setup of using a non-linear neural network base ODE, both the INN and neural network dynamics are trained jointly. We empirically investigate the representation burden on the invertible neural network, when using a non-linear neural network base ODE. We train, for 5000 epochs, augmented ODEs with neural network dynamics models of different parameter sizes, without the diffeomorphism, on the Lorentz system and report the MAE values of integrating with the dopri5 integrator with the same setup as the Lorentz experiments in Section 5.2. We then compare take the neural network dynamics model with the smallest number of parameters (with a parameter count of 2256), and add INNs with an increasing number of invertible layers. The results of the ODE with neural network parameters only, with increasing parameter counts, are tabulated in table 3, while the results of the neural networks with an increasing number of invertible layers are tabulated in table 4.

Each invertible layer contains 15090 parameters. We note here that the querying the INN (and thereby having a large INN) adds negligible integration time, as during the entire integration we only need one forward and one inverse pass of the INN, while the system dynamics needs to be queried sequentially. We see that, in general, adding parameters to the system dynamics of a learned ODE and to the diffeomorphism adds to the representation power of our ODE models. However, we clearly see that the addition of an INN improves performance, and additional layers to the INN, up to around 5 invertible layers, improves performance. We see that the model with 5 invertible layers and a base ODE of 2256 parameters greatly outperforms the ODE model with only a neural network dynamics of 83006, while the two models are similar in parameter count. This indicates that the invertible neural network plays a large role in representing the learned ODE.

| NN-dynamics parameter count | 2256 | 8106 | 17556 | 30606 | 47256 | 67506 | 83006 |
|---|---|---|---|---|---|---|---|
| MAE | 10.95 | 10.94 | 10.94 | 8.04 | 7.35 | 7.06 | 7.08 |

Table 3: The MAE of interpolating trajectories from the Lorentz system, with ODEs using neural networks of various sizes.

| Number of INN Layers | 0 | 1 | 2 | 3 | 4 | 5 | 6 | 7 |
|---|---|---|---|---|---|---|---|---|
| MAE | 10.95 | 2.9 | 2.53 | 0.76 | 0.45 | 0.21 | 0.38 | 0.29 |

Table 4: The MAE of interpolating trajectories from the Lorentz system, with a small base ODE with 2256 parameters and an increasing number of invertible layers.

## B    ADDITIONAL RESULTS: TRAINING TIMES

We present the training times for directly learning ODEs with our method, using a linear base ODE. These include the training times on the 3D Lotka-Volterra, and the robot imitation datasets, outlined in Sections 5.1.1 and 5.1.2 of the main paper. We run training for 1000 iterations, where in each iteration the batch includes the entire training set. We see that, by leveraging the closed-form solution of linear ODEs, our method is able to also drastically speed up training. Additionally the parallel nature of passing through the invertible neural network allows more consistent training times across datasets.

| | 3D Lotka-Volterra | | Imitation S | | Imitation cube pick | | Imitation C | |
|---|---|---|---|---|---|---|---|---|
| | Per iter (s) | Total (s) | Per iter (s) | Total (s) | Per iter (s) | Total (s) | Per iter (s) | Total (s) |
| Ours | 0.030± 0.002 | 29.57 | 0.029± 0.002 | 29.35 | 0.031± 0.007 | 31.34 | 0.030± 0.004 | 29.78 |
| Euler | 0.131±0.003 | 130.94 | 1.740± 0.017 | 1740.43 | 1.713± 0.016 | 1712.53 | 1.704± 0.010 | 1703.85 |
| Midpoint | 0.235±0.006 | 235.07 | 3.228± 0.027 | 3227.51 | 3.177± 0.028 | 3177.37 | 3.207± 0.025 | 3206.92 |
| RK4 | 0.469±0.005 | 468.52 | 6.671± 0.048 | 6671.44 | 6.388± 0.046 | 6388.10 | 6.441± 0.057 | 6440.61 |
| Dopri5 | 0.408±0.037 | 408.36 | 1.413± 0.034 | 1413.34 | 1.246± 0.023 | 1245.65 | 1.247± 0.022 | 1246.67 |

Table 5: The training times in seconds with standard deviations, for 1000 iterations. By leveraging the closed-form solution of linear ODEs, training time with our method is consistently orders of magnitude faster than by using a differentiable numerical integrator.

## C   ADDITIONAL RESULTS: ABLATION STUDY

We study the effects of the number of layers in the invertible neural network and number of parameters in the sub-network, which are the main hyper-parameters of the invertible neural networks used. To this end, we conduct ablation studies of the speed and performance of our method on the real-world datasets outlined in section 5.1.2 of the paper. Our basic model uses an invertible neural network with 5 layers and sub-networks in the invertible network had 1500 hidden dimension size. We alter the number of layers to be: 2, 3, 4, 5, 6, 7, 8, and hidden dimensions of the sub-networks within the invertible network to be: 500, 1000, 1500, 2000, 2500. The results are presented below:

| | | Imitation S | | Imitation cube pick | | Imitation C | |
|---|---|---|---|---|---|---|---|
| No. Layers | Sub-Net Hid. Dim. Size | Int. time (ms) | MSE | Int. time (ms) | MSE | Int. time (ms) | MSE |
| 2 | 1500 | 3.551± 0.585 | 122.40 | 2.993±0.103 | 41.51 | 2.914±0.049 | 20.69 |
| 3 | 1500 | 4.589± 1.193 | 130.49 | 4.026±0.129 | 15.00 | 5.150±1.407 | 26.20 |
| 4 | 1500 | 5.418± 0.746 | 24.54 | 6.294±0.939 | 26.18 | 5.374±0.382 | 10.33 |
| 5 | 1500 | 6.461± 0.686 | 4.40 | 7.401±1.531 | 26.56 | 7.463±1.929 | 13.16 |
| 6 | 1500 | 7.529± 0.698 | 8.17 | 8.993±2.212 | 17.16 | 8.994±2.781 | 18.27 |
| 7 | 1500 | 9.426± 1.664 | 4.91 | 9.858±1.828 | 20.39 | 9.669±1.240 | 25.76 |
| 8 | 1500 | 10.636± 2.541 | 5.62 | 10.732±2.111 | 14.56 | 10.958±2.518 | 6.57 |
| 5 | 500 | 7.159± 1.475 | 5.62 | 8.018±2.109 | 19.37 | 7.315±1.483 | 6.22 |
| 5 | 1000 | 6.972± 1.247 | 6.04 | 6.376±0.203 | 11.02 | 7.297±1.311 | 6.37 |
| 5 | 1500 | 7.031± 1.289 | 4.40 | 7.321±1.236 | 26.56 | 6.901±0.802 | 13.16 |
| 5 | 2000 | 7.787± 1.363 | 6.23 | 7.443±1.457 | 14.05 | 7.776±2.514 | 9.48 |
| 5 | 2500 | 7.208± 1.628 | 10.92 | 6.521±0.082 | 11.66 | 7.611±1.687 | 5.59 |

Table 6: Ablation study results of different configurations for the invertible neural network model.

We see that as we increase the number of invertible network layers, the integration times increase, while the hidden dimension size of the sub-networks within the invertible network does not visibly affect the integration times. Overall, the generalisation performance improves as the number of invertible layers are used, up to some number of layers. Beyond this number of layers, adding layers does not vary performance significantly. Additionally, the hidden dimension sizes, for the values tested do not greatly vary the generalisation performance.

## D   PROOFS

Proofs for Propositions 4.1 and 4.2 can be found in Lee (2012) as Propositions 8.19 and 9.6.

**Theorem 4.1.** *Suppose two ODEs* $\mathbf{x}'(t) = g(\mathbf{x}(t))$, $\mathbf{y}'(t) = f(\mathbf{y}(t))$ *are related via* $\mathbf{y}(t) = F(\mathbf{x}(t))$, *where* $F$ *is a diffeomorphism. If the former ODE is asymptotically stable with* $n_e$ *equilibrium points* $\mathbf{x}_1^*, \ldots, \mathbf{x}_{n_e}^*$, *then the latter ODE is also asymptotically stable, with equilibrium points* $F(\mathbf{x}_1^*), \ldots, F(\mathbf{x}_{n_e}^*)$.

*Proof.* First we show $F(\mathbf{x}_1^*), \ldots, F(\mathbf{x}_{n_e}^*)$ are equilibrium points of ODE $\mathbf{y}'(t) = f(\mathbf{y}(t))$. By $\mathbf{y}(t) = F(\mathbf{x}(t))$, we can write the time derivatives $\mathbf{y}'$ at $F(\mathbf{x})$ as

$$\mathbf{y}'(t) = f(F(\mathbf{x}(t))) = \frac{\mathrm{d}F(\mathbf{x}(t))}{\mathrm{d}t} = J_F(\mathbf{x}(t))g(\mathbf{x}(t)), \tag{8}$$

where $J_{F(\mathbf{x}(t))}$ is the Jacobian of $F$. $F$ is a diffeomorphism and hence invertible over its domain. By the inverse function theorem (Dontchev & Rockafellar, 2009), the Jacobian $J_F(\mathbf{x}(t))$ is invertible, and furthermore, by the invertible matrix theorem (Horn & Johnson, 2012), it has a null-space containing only the zero vector. Therefore, $\mathbf{y}'(t) = f(F(\mathbf{x}(t))) = J_F(\mathbf{x}(t))g(\mathbf{x}(t)) = 0$ if and only if $g(\mathbf{x}(t)) = 0$. As $g(\mathbf{x}^*(t)) = 0$ for $\mathbf{x}^* \in \{\mathbf{x}_1^* \ldots \mathbf{x}_{n_e}^*\}$, then we also have $f(F(\mathbf{x}^*(t))) = 0$, hence $\mathbf{y}^* \in \{F(\mathbf{x}_1^*), \ldots, F(\mathbf{x}_{n_e}^*)\}$ gives equilibrium points for $\mathbf{y}'(t) = f(\mathbf{y}(t))$.

We now show asymptotically stability of $\mathbf{y}'(t) = f(\mathbf{y}(t))$, by the existence of a *Lyapunov function* (Lefschetz & Alverson, 1962), $V_{\mathbf{y}} : \mathbb{R}^n \to \mathbb{R}$, where $n$ is the dimension of $\mathbf{y}$, such that $\frac{\partial V_{\mathbf{y}}(\mathbf{y})}{\partial t} < 0$ for all $\mathbf{y} \in \mathbb{R}^n \setminus \{F(\mathbf{x}_1^*), \ldots, F(\mathbf{x}_{n_e}^*)\}$, and $\frac{\partial V_{\mathbf{y}}(\mathbf{y}^*)}{\partial t} = 0$ for $\mathbf{y}^* \in \{F(\mathbf{x}_1^*), \ldots, F(\mathbf{x}_{n_e}^*)\}$. The existence of such a Lyapunov function is a necessary and sufficient condition for stability. We assume the candidate function to be $V_{\mathbf{y}} = V_{\mathbf{x}}(F^{-1}(\mathbf{y}))$, where $V_{\mathbf{x}}$ is a valid Lyapunov function of the asymptotically stable $\mathbf{x}'(t) = g(\mathbf{x}(t))$, with $\frac{\partial V_{\mathbf{x}}(\mathbf{x})}{\partial t} < 0$ for $\mathbf{x} \in \mathbb{R}^b \setminus \{\mathbf{x}_1^*, \ldots, \mathbf{x}_{n_e}^*\}$ and $\frac{\partial V_{\mathbf{x}}(\mathbf{x}^*)}{\partial t} = 0$ for $\mathbf{x}^* \in \{\mathbf{x}_1^*, \ldots, \mathbf{x}_{n_e}^*\}$. Consider the time derivative of the candidate function:

$$\frac{\partial V_{\mathbf{y}}(\mathbf{y})}{\partial t} = \frac{\partial V_{\mathbf{y}}}{\partial \mathbf{y}}\frac{\partial \mathbf{y}}{\partial t} = \frac{\partial V_{\mathbf{y}}}{\partial \mathbf{y}}f(\mathbf{y}) \tag{9}$$

$$= \left(\frac{\partial V_{\mathbf{x}}}{\partial \mathbf{x}}\frac{\partial F^{-1}}{\partial \mathbf{y}}\frac{\partial F}{\partial \mathbf{x}}g(\mathbf{x})\right)_{\mathbf{x}=F^{-1}(\mathbf{y})} \tag{10}$$

$$= \left(\frac{\partial V_{\mathbf{x}}}{\partial \mathbf{x}}J_F(\mathbf{x})^{-1}J_F(\mathbf{x})g(\mathbf{x})\right)_{\mathbf{x}=F^{-1}(\mathbf{y})} \tag{11}$$

$$= \left(\frac{\partial V_{\mathbf{x}}}{\partial \mathbf{x}}g(\mathbf{x})\right)_{\mathbf{x}=F^{-1}(\mathbf{y})} = \left(\frac{\partial V_{\mathbf{x}}(\mathbf{x})}{\partial t}\right)_{\mathbf{x}=F^{-1}(\mathbf{y})}. \tag{12}$$

Equation (11) by the inverse function theorem (Dontchev & Rockafellar, 2009). Therefore, our candidate $V_{\mathbf{y}}$ is a valid Lyapunov function for $\mathbf{y}'(t) = f(\mathbf{y}(t))$. Thus, the system $\mathbf{y}'(t) = f(\mathbf{y}(t))$ is asymptotically stable. $\qquad\square$

# E  ADDITIONAL IMPLEMENTATION DETAILS

We run all of our experiments on a machine with an Intel i7-3770k 3.50GHz processor, 32GB RAM and an NVIDIA GTX1080 GPU, with 8GB vRAM. For all of our experiments, we use the optimizer ADAM with step-size $10^{-4}$, except for the experiments in the Latent ODE, which where we use the standard set-up from the Latent-ODE repository (Rubanova et al., 2019). The dynamics models of compared ODEs have the architecture: Input->dense(Input dimensions, 150)->tanh()->dense(150,150)->tanh()->dense(150,150)->tanh()->dense(150,150)->tanh()->dense(150,150)->tanh()->dense(150,output dimensions)->output. Except for the Latent-ODE comparisons where settings from the original repository (Rubanova et al., 2019) is used. For all of the experiments, except latent ODE experiments where we follow the original set-up, we train for 5000 iterations in total.

For all the experiments where we directly learn a dynamical system, we use an invertible neural network with 5 invertible layers, and sub-networks with one hidden layer of 1500 units. For non-linear base ODEs parameterized with a simple neural networks, we use the architecture: Input->dense(Input dimensions,30)->tanh()->dense(30,30)->tanh()->dense(30,30)->tanh()->dense(30,Output dimensions)->Outputs. Additionally, all learned dynamics, both with ours and compared methods, excepted when adhering to the original Latent-ODE set-up, were augmented with the same number of additional zeros as original state dimensions, for example 3 dimensional systems were augmented to 6 dimensions. An exception to this is the high-dimensional image rotation problem, where we found that adding half as many augmented states as the original state dimensions was sufficient.

The Lotka-Volterra system used has the dynamics:
$$x'(t) = x(t)(0.75 - 0.75y(t)) \tag{13}$$
$$y'(t) = y(t)(-0.75 + 0.75x(t) - 0.75z(t)) \tag{14}$$
$$z'(t) = z(t)(-0.75 + 0.75y(t)) \tag{15}$$

for $t \in [0, 7]$ with initial conditions $\{(5, 5, 1), (2, 6, 6), (3, 1, 4), (7, 1, 2), (6, 2, 4),$
$(3, 3, 1), (2, 2, 2), (4, 4, 3), (3, 3, 4), (1, 1, 5)\}$.

The Lorenz system used has the dynamics:

$$x'(t) = 10(y(t) - x(t)) \tag{16}$$
$$y'(t) = x(t)(28 - y(t)) - x(t) \tag{17}$$
$$z'(t) = x(t)y(t) - \frac{8}{3}z(t) \tag{18}$$

for $t \in [0, 2]$ with the initial conditions $(0.15, 0.15, 0.15)$.

The Robertson's system used has the dynamics:

$$x'(t) = -0.04x(t) + 3 \times 10^4 y(t)z(t) \tag{19}$$
$$y'(t) = 0.04x(t) - 3 \times 10^4 y(t)^2 - 10^4 y(t)z(t) \tag{20}$$
$$z'(t) = 3 \times 10^4 y(t)^2 \tag{21}$$

for $t \in [0, 120]$ with the initial conditions $(1, 0, 0)$. During training and evaluation, we rescale the data dimensions, and roll out the ODE in log-space.

In the latent-ODE problem setup, an observable time-series is assumed to have latent variables which follow some ODE dynamics, and uses an `Encoder -> ODE -> Decoder` architecture where an ODE is used to model the hidden latent dimensions between the Encoder and Decoder. Note that a valid ODE is not guaranteed in the space of observable data, but only in the latent dimensions. Our set-up follows the repository given by Rubanova et al. (2019), with the training settings for the Encoder and Decoder architecture as below:

**Periodic 100**: We train the entire model for 500 epochs with Adamax optimizer and an initial learning rate of $10^{-2}$. We sub-sample 5% of the original time points and the size of the latent state is 10. The noise weight is set as 0.01 and the total number of time points is 100. For the Neural ODE architectures, there is one layer in the recognition ODE and one layer in the generative ODE, and 100 unit per layers. For the GRU unit there exists 100 units per layer for the GRU update network. All the above settings are exactly the same as the configuration given in repository (Rubanova et al., 2019).

**Periodic 1000**: Settings are the same as *Periodic 100*, except that the total number of time points is set as 1000 to predict for finer time steps.

**Human Activity**: The model is trained for 200 epochs, with a dimensionality of 15 in the latent state. There are 4 layers in the recognition ODE and 2 layers in the generative ODE, and 500 units per layer. The GRU unit has exists 50 units per layer.

**ECG**: Settings are the same as the classification task of *Human Activity*, except that we use the ECG Heartbeat data available at Dataset (2018), removing the '0' class.

**Rotating Image Sequence**: We use the first 100 MNIST "3" characters as training and the next "3" characters as test. We create a sequence of 44 images for each initial character until we rotate the initial image by 180 degrees. During testing, we integrate to obtain a sequence of 440, at 10x data resolution. We obtain latent representations of each image by training a convolutional autoencoder to obtain a 64 dimension latent vector. We fit the ODE on these latent dimensions, with an ADAM optimizer with learning rate $5 \times 10^{-4}$.

**ODE dimensions and sequence lengths**: The following table contains details on the dimensions of the ODE models. For latent ODE models, these are the dimensions of the latent state. We also provide the lengths of the integrated trajectories of the ODEs during inference.

## F    ADDITIONAL FIGURES

We provide figures for learning an additional Lorenz system for $t \in [0, 5]$, with trajectory at initial condition $(-3.1, -1.15, 8.15)$. We see that our method, with a base ODE parameterized by a neural network, can generate trajectories that closely match the ground truth:

| | Dimensions of ODE to Learn | Time points in Trajectory |
|---|---|---|
| 3d-LV | 3 | 700 |
| Imitate S | 2 | 1000 |
| Pick cube | 3 | 1000 |
| Imitate C | 3 | 1000 |
| Periodic 100 | 10 | 100 |
| Periodic 1000 | 10 | 1000 |
| Human Activity | 15 | 157 |
| ECG | 15 | 188 |
| Rotating MNIST | 64 | 440 |
| Lorenz | 3 | 800 |
| Robertson | 3 | 500 |

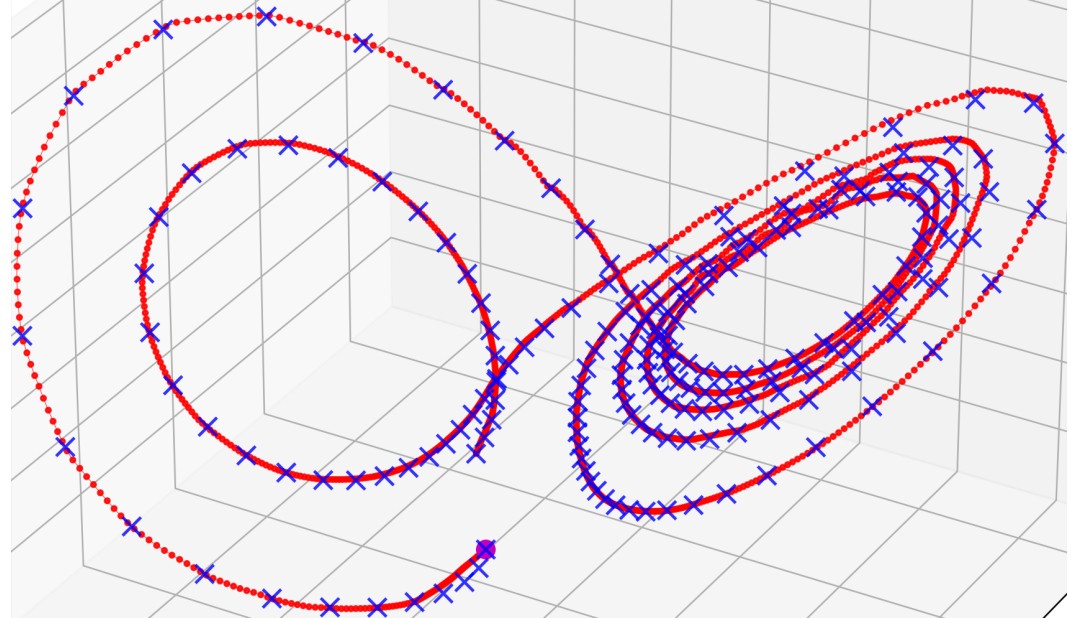

Figure 6: A learned Lorenz system with the generated trajectory, at 10x data resolution, and ground truth.

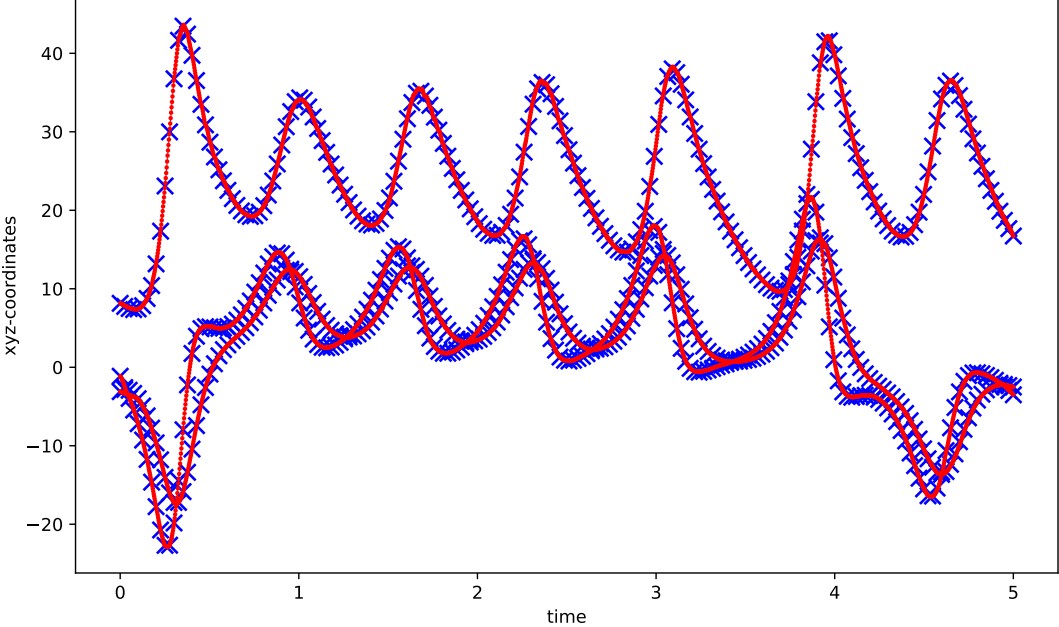

Figure 7: A learned Lorenz system with the generated trajectory, at 10x data resolution, and ground truth, rolled out in time

We provide the change in coordinates over time, for the trajectory shown in figure 5(a) in the paper:

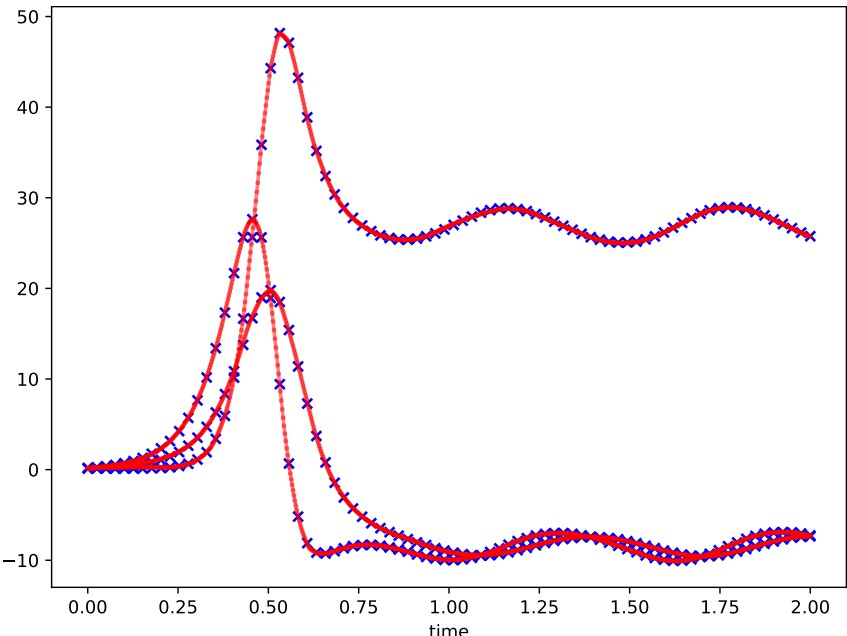

Figure 8: The corresponding plot showing the coordinates over the time interval of a learned Lorenz system over $t \in [0, 2]$, at 10x data resolution, which corresponds to the 3d figure shown as fig 5 (left).

We provide additional plots of trajectories, at different start points, from a learned Lotka-Volterra system. The ground truth data is in blue, while generated trajectory, of 10x data resolution, is in red.

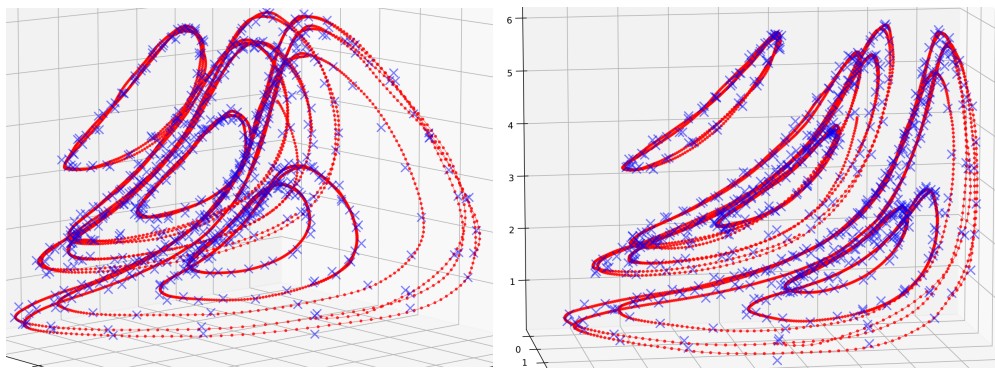

Figure 9: We see that trajectories from the learned Lotka-Volterra system, in red, closely matches the ground truth, in blue.

We provide an additional figure for trajectories generated at unseen starting points after being trained on the "imitation C" training data. The four generated trajectories are in red, while the ground truths are in blue. Our generated trajectories match the ground truth, and accurately capture the motion of drawing a "C" character.

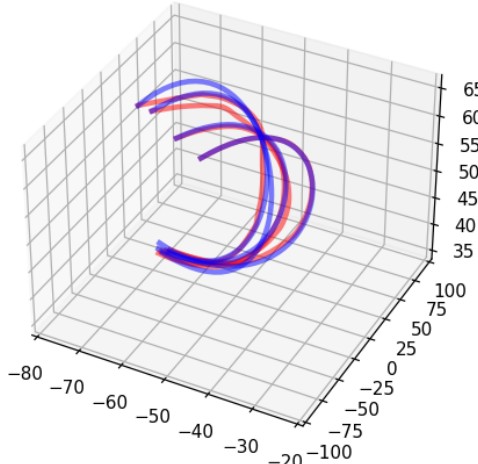

Figure 10: Robot motion trajectories in red, that imitate drawing a "C" character. The ground truth is given in blue.

## G    LICENSES FOR PACKAGES

Common scientific packages used in our code include: (i) Numpy (Harris et al., 2020) (BSD license), for general linear algebra and miscellaneous math operations (ii) Matplotlib (Hunter, 2007) (BSD compatible custom license), for plotting figures.

More specialized packages used include (i) FrEIA (Ardizzone et al., 2019) (MIT license), for invertible neural networks; (ii) TorchDiffEq (Chen et al., 2018) (MIT license), for differentiable numerical integrators; (iii) Pytorch (Paszke et al., 2019) (BSD license), for optimisation and automatic differentiation; (iv) Latent-ODE (Rubanova et al., 2019) (MIT license), for latent ODE implementation.

