# OpenReview forum: "Learning Efficient and Robust Ordinary Differential Equations via Diffeomorphisms"
_ICLR.cc/2022/Conference — ICLR 2022 Submitted_

### Official Review · Reviewer_HKf9 · 2021-10-25

**Correctness:** 2
**Technical Novelty And Significance:** 2
**Empirical Novelty And Significance:** 2
**Recommendation:** 3
**Confidence:** 4

**Main Review:**

Learning an extra coordinate transformation is a very interesting idea and shared by important recent works. It is also closely related to encoder and decoder (although this is not discussed). I feel the authors have a great idea that is definitely worth pursuing, although the paper in its current form could use some significant improvements.

First of all, putting linear and NODE-base-vector-field together appears a little unnatural to me. Is a linear base recommended or not? I would appreciate it more if they could be argued to be complementary, and a criterion is given as such when to use which. Here are some of my understandings of their difference in scopes and applicability:

Let me first discuss the linear base. In this case, despite of the extra differomorphism, the topology of the learned dynamics is still going to be linear. This is very restricted. Topological conjugacy can change the quantitative appearance of dynamics but not its structures, such as the topology of limit sets. All examples in 5.1 reflect this fact. Can the author use linear base + diffeomorphism to capture both the libration and circulation regions of a pendulum, as well as the separatrix inbetween? If not, I think it is important not to exaggerate the representation power of linear bases.

An addition comment on the linear base case is, the stability criterion for autonomous linear systems is well-known, and stability is nontrivial only for truly nonlinear problems. I would rather see more details of the core treatments and discussions, instead of loose parts that try to make the paper appear more mathematical (e.g., Definition 4.2, 4.3, Theorem 4.1, and all the parts involving manifolds, e.g., 3.2, Figure 2, Definition 4.1, Proposition 4.1, as the work doesn’t really have a manifold component).

The second approach, `non-linear neural network base’, should work for general problems though. This base is not new, and the contribution is to add a coordinate transform layer. For this part, it would be great if two matters are better discussed: (1) uniqueness. Now both the differmorphism and the base dynamics are learned, and given one differmorphism there is a base dynamics. Which one will be chosen? There will be implicit bias imposed by both the architecture and the training methodology. I think theoretically understanding this would be challenging, but at least some empirical investigation would be helpful. (2) Why is this better than just learning the vector field? The authors did mention on page 5 that this is computationally more advantageous, but I’m sorry that I’m not very convinced.
In addition, I’d like to better understand how the new method scales with the dimension. As this paper proposes an improvement of NODE, comparisons with NODE-based approaches on machine learning baselines would be very helpful.

Additional points:
* It seems the Lorenz example in Fig.5 is not for a chaotic trajectory, so it diffuses the point of testing on a Lorenz system.
* Some important NODE references are given, but the expansive data-driven learning differential equation literature is almost completely missing.
* Eq.(2): $ODESolve(f_\omega, y_{t_0}, t_e)$ should be $ODESolve(f_\omega, y_{t_0}, t_0, t_e)$ or $ODESolve(f_\omega, y_{t_0}, t_e-t_0)$.
* Right before eq.(2): Euler’s method is a Runge-Kutta method.


**Summary Of The Paper:**

In order to better learn differential equations that produce discrete data, this paper proposes to learn a coordinate transform (i.e. the `diffeomorphism’ in the title) in addition to learning the vector field. It is argued that, after this coordinate transformation, one can either learn a linear ODE or learn a neural network parameterized ODE, and obtain improved efficiency and robustness.

**Summary Of The Review:**

The "linear base" part and the "non-linear neural network base" part are disjoint. Unlike the current wordings imply, I think the former is rather restrictive. On the other hand, the latter is interesting but needs more work. Although I cannot recommend the acceptance of this paper in its current form, I encourage the authors to strengthen the latter part and consult more with the literature.

---

> ### Author Response · Authors · 2021-11-19
> **Response to Reviewer HKf9**
>
> We would like to thank the reviewer for the helpful feedback, and the appreciation of the ideas we have presented. Here, we’d like to clarify and respond to some of the comments.
>
> > Representation power of the linear ODE base + diffeomorphism
>
> We acknowledge that this setup has limitations in that the bijection does not change the topological structure of the base ODE. The same criticism of bijectivity appears in the normalizing flows literature [1]. We note that for many typical ODE learning setups the structure of the desired ODE can be captured by a linear base, as demonstrated in our experiments. For example, in problems of learning and generating robot motions in the learning for demonstration set-up, a classic problem is learning a dynamic system to generate  drawing “S” characters from different start points. A linear ODE cannot be fit to learn such a system, yet a linear base + diffeomorphism can represent such a system. We note that this set-up provides huge speed improvements compared to learning the ODE with a differentiable integrator + neural network dynamics. We present both the linear ODE and the more general non-linear ODE base as complementary, and not as substitutes of one another.
>
> > ”An addition comment on the linear base case is, the stability criterion for autonomous linear systems is well-known”
>
> Following from the last point, using a linear base ODE + diffeomorphism is able to learn dynamics which cannot be represented by a linear ODE alone. By constraining the base ODE to be stable, we can enforce stability on the ODE + diffeomorphism.
>
> > “Why is this better than just learning the vector field”
>
> In our experiments, we show that neural network base + bijection outperform simply using an ODE with neural network system dynamics with even more parameters, when the system is difficult to learn and contains rapid variations. This is closely related to the next point:
>
> > "I think theoretically understanding this would be challenging, but at least some empirical investigation would be helpful. "
>
> To investigate the load on the invertible neural network, we have added a new section in the appendices which investigates the effect of increasing the number of invertible layers (thereby the flexibility of the invertible network), relative to simply increasing the parameter size of a neural network dynamics model, when learning the Lorentz system.
>
>
> > “The authors did mention on page 5 that this is computationally more advantageous”
>
> The speed improvements here come from the fact that our system dynamics model of our base ODE can have much fewer parameters, than the system dynamics model required to directly learn the ODE. The pass through the learned bijection can be done very quickly and be batched to run on a GPU, whereas the numerical integrator requires sequentials queries to the “heavier” dynamics model. This is shown in the performance of our model against comparisons in section 5.2.
>
> >"In addition, I’d like to better understand how the new method scales with the dimension. "
>
> Our method is able to learn high-dimensional ODE systems. The dimensions of all our experiments are shown in the table in section D of the appendices. We would like to highlight here that our experiments include systems of ODEs which are of 64 dimensions.
>
> > "As this paper proposes an improvement of NODE, comparisons with NODE-based approaches on machine learning baselines would be very helpful."
>
> We would like to point to the experiments in "ODE Learning for Continuous Deep Learning Models" of section 5.1. We experiment with many of the on machine learning baseline problems which appear in Rubanova et al., 2019, these include the time series prediction problems of human activity classification and ECG classification.
>
> > "It seems the Lorenz example in Fig.5 is not for a chaotic trajectory"
>
> Section E in the appendices gives more examples of trajectories from the Lorentz which are chaotic.

---

### Official Review · Reviewer_9Lrk · 2021-10-31

**Correctness:** 4
**Technical Novelty And Significance:** 4
**Empirical Novelty And Significance:** 2
**Recommendation:** 6
**Confidence:** 5

**Main Review:**

The paper addresses a very timely topic and shows a very interesting connection between differential geometry and ODE systems. I find this theoretical contribution (or rather perspective) highly significant. Considering the convincing experimental results, I believe the paper should be accepted. For the first round of reviewing, I'm giving a score of 6 but would happily increase it if my following concerns are addressed:

- The connection with the latent neural ODEs is missing. One could even present the proposed methodology as a special case of latent neural ODEs; therefore, showing and justifying the differences are important. Similarly, I would like to see empirical comparisons with latent neural ODEs to be written more clearly. As such, all tables are presented as if the competing method is the same across tables.
- I'm not sure if the informal definition of "stiff ODEs" in Sec 4.6. is correct. No my knowledge, a stiff ODE system does not necessarily rapidly vary along a dimension but rather causes numerical solvers to be stuck at certain points in the space.
- Are the number of model parameters comparable? Since the presented approach utilizes two networks, I believe the comparison should be done fairly. This is questionable looking at Sec.D. Similarly, it is questionable that the adaptive step solvers are used with correct tolerance values.
- What are the initial values for integration? This is mentioned nowhere in the main paper. Also, the effect of the noise on the initial values should be discussed.
- I wonder how much of the "load" is on the shoulders of the base ODE system. This might not be of vital importance but "disentangling" the contributions of the invertible layer and the ODE system might be beneficial.

Minor comments:
- The paper is written clearly. I particularly enjoyed the preliminaries section and how the concepts are explained in sufficient detail. That being said, the writing can be improved:
  - Section 4.3 is a little repetitive. In particular, the third and fourth paragraph partially repeat the information above.
  - To my knowledge, he phrase "integral curve" is not commonly used in neural ODE community. Giving its formal definition or synonyms would help the reader.
  - Section 4.4 is highly repetitive (can be even completely removed). Also, some title along the line "summary of our methodology" could be better since the word "result" typically refers to experimental findings.
  - The last sentence above Def. 4.2 reads strange. I would suggest authors revisit this paragraph for coherence.
- Figure 3 seems extremely crowded (especially the right one).
- The long sentences (especially in the table captions and the appendix) should be revised.
- The last paragraph in page 8 is extremely long and requires a better structure.
- It is a little difficult to see how significant the differences in Table1 are. It would be nice to see a visualization of forward trajectories over time.

**Summary Of The Paper:**

This paper presents an interesting perspective that connects ordinary differential equations with diffeomorphisms. More specifically, possibly complicated and thus difficult-to-learn ODE dynamics are expressed as a morphed version of a simple "base" ODE system. The authors propose to learn the base dynamics alongside with the corresponding diffeomorphism. As experimentally shown, this approach leads to faster training, somewhat increases accuracy and allows for the long-term behaviour analysis of certain types of base ODE systems.

**Summary Of The Review:**

Since the paper shows interesting links between seemingly unrelated topics, I find the contribution significant and suggest an accept although the writing can be improved and the model can be investigated in more detail.

---

> ### Author Response · Authors · 2021-11-23
> **Response to Reviewer 9Lrk**
>
> We would like to thank the reviewer for the helpful feedback, and the appreciation of the perspectives we have presented. Here, we’d like to clarify and respond to some of the comments.
>
> > “The connection with the latent neural ODEs is missing. One could even present the proposed methodology as a special case of latent neural ODEs”
>
> Although in spirit the structure of our model is similar to latent ODEs, our method is different in a crucial way: our method produces an ODE in the input space of the data, while latent ODEs do not produce a valid vector field in the input space, but the vector field is rather defined in the latent space. This is important for specific problems in robot learning, such as that introduced in the introduction section of [2], where we wish to obtain a valid ODE to represent the dynamics. We have now added a clarification of this in section 4.1.
>
> > “I'm not sure if the informal definition of "stiff ODEs" in Sec 4.6. is correct. a stiff ODE system does not necessarily rapidly vary along a dimension but rather causes numerical solvers to be stuck at certain points in the space.”
>
> A stiff equation is a differential equation for which numerical solvers tend to get stuck, unless the step size is taken to be extremely small. It has proven difficult to formulate a precise definition of stiffness, but the main idea is that the equation includes some terms that can lead to rapid variation in the solution, resulting in the solver getting stuck. [1]
>
> > “Are the number of model parameters comparable? Since the presented approach utilizes two networks, I believe the comparison should be done fairly. This is questionable looking at Sec.D. Similarly, it is questionable that the adaptive step solvers are used with correct tolerance values. ”
>
> Although the invertible neural network may have a fairly large number of parameters, the contribution of the INN to integration time is fairly negligible when using a neural network base ODE. During integration, we perform one forward and one inverse pass with the INN, whereas the neural network dynamics needs to be queried sequentially. The adaptive solvers had absolute and relative tolerances set at 1e-5, this is a fairly standard tolerance.
>
> For more information on the comparisons, we have added a new section in the appendices tabulating the performance of different neural network dynamics models with different numbers of parameters. We have also looked at the effect of adding a different number of invertible layers.
>
> > “What are the initial values for integration?”
>
> The initial values for integration for the various tested systems are outlined in sec. E of the appendices, due space constraints, these did not appear in the main paper.
>
> > “"disentangling" the contributions of the invertible layer and the ODE system might be beneficial. ”
>
> We provide additional empirical results in section A of the appendices, where we add an increasing number of invertible layers to a simple base ODE, on the Lorentz system experiment . We find that there is a very significant increase in performance when invertible layers are added, up to a certain limit. This amount of improvement in performance cannot be achieved by increasing the parameter count of the network modelling the ODE dynamics, indicating a significant load is on the invertible neural network.
>
>
> [1] Hairer, Ernst; Wanner, Gerhard (1996), Solving ordinary differential equations II: Stiff and differential-algebraic problems (second ed.),
> [2] “Learning Contracting Vector Fields For Stable Imitation Learning”, Sindhwani et al, 2018. https://arxiv.org/pdf/1804.04878.pdf.

---

### Official Review · Reviewer_HKa2 · 2021-11-02

**Correctness:** 3
**Technical Novelty And Significance:** 2
**Empirical Novelty And Significance:** 2
**Recommendation:** 3
**Confidence:** 4

**Main Review:**

I don’t agree with the dichotomy presented in paragraph 1 of the related work:

“Our proposed approach improves the learning of the underlying ODE, and is compatible with models that incorporate learnable ODEs. We note that the term “neural ODE” has typically been used in the literature to refer to neural networks that incorporate ODEs, including the original work in Chen et al. (2018). However, “neural ODE” has occasionally been used to refer to an ODE with dynamics parameterized by a neural network (Norcliffe et al., 2021). We follow the former convention.”

My understanding is “Neural ODE” refers to ODEs parameterized by neural networks. This includes the case where such an ODE is incorporated into a deep learning model. See the second paragraph of [1]

Related work: I think [1], [2], [3], [4] should be included in the related work section, as they all propose methods to improve the efficiency of neural ODEs. [4] is very recent, so maybe it’s not necessary to cite it, but it seems very related to this paper.

Can you more thoroughly explain the benefits of asymptotically stable Neural ODEs? How do these models maintain expressivity while being asymptotically stable? Are there example of Neural ODEs without diffeomorphisms that can be empirically demonstrated to be asymptotically unstable?

It doesn’t seem that this definition of asymptotic stability in Definition 4.3 permits periodic solutions, yet there are experiments with these dynamics. Can you explain this?

A principal advantage of composing a diffeomorphism with an ODE, as outlined in Section 4.3, rather than an arbitrary function, is that solution curves can be obtained without evaluating the jacobian of the diffeomorphism. I’m not sure why this is a big advantage.
- Multiplying by the Jacobian of the diffeomorphism is relatively cheap (2x more expensive than evaluating F) via vector-jacobian products. Do quantities besides a vector product need to be computed (e.g. the trace of the jacobian?)
- If we replaced F with some non-invertible neural network, we would also not require evaluating its jacobian to compute predictions from the model. We would need two separate neural networks (for pre- and post-processing, respectively) to match the expressivity, but otherwise I don’t see why we need to maintain the invertibility beyond the fact that we’d like the entire function to be an ODE.

How do we make use of the fact that the entire model is one well-defined vector field / ODE? The principal advantage seems to be for some time series problems, where the whole function is now an ODE.

Why is there a requirement that the model be entirely composed of one ODE? Many neural ODE models incorporate preprocessing steps, and the pre- and post- processing steps are not inverses of each other (e.g. one might be downsampling). Training continuous normalizing flows might be one application where a diffeomorphism is required, and experiments on this task would be interesting.

It isn’t well-characterized when it is suitable to use a Linear Base ODE vs. a neural network ODE. When is the linear base ODE expressive enough? Are there tractable components that can be composed with sufficient expressivity, instead of having to resort to a neural network ODE?

Are there more fine-grained measures of time-cost of different models? For example, function evaluations from the numerical integrator can be measured as a more robust measure than the total seconds elapsed. Does Table 2 include results from models in Rubanova et al. 2019? Or is it just Linear ODE models integrated with different solvers?

The experiments are not extensive enough. Perhaps it might be worth trying some of the tasks investigated in [4]?

Typos:
Section 2:
- “normlizing flows” (second paragraph)
Section 4.1:
- “If mapping F” (second paragraph)
Section 4.2:
- “and get the inverse with no further labor” (first paragraph)

[1]: “Learning Differential Equations that are Easy to Solve”, Kelly et al. (arxiv.org/abs/2007.04504)
[2]: “How to train your Neural ODE”, Finlay et al. (​​arxiv.org/abs/2002.02798)
[3]: “Opening the Black Box: Accelerating Neural Differential Equations by Regularizing Internal Solver Heuristics”, Pal et al. (arxiv.org/abs/2105.03918)
[4]: “Neural Flows: Efficient Alternative to Neural ODEs”, Biloš et al. (arxiv.org/abs/2110.13040)


**Summary Of The Paper:**

ODEs parameterized by neural networks can be slow when learning systems from long sequences. This paper proposes learning an ODE with a fast diffeomorphism, so that the ODE can be a simpler function that’s more efficient to numerically integrate, while preserving expressivity.

**Summary Of The Review:**

The paper does not explain clearly enough the benefits of the proposed approach over regular Neural ODE models, nor thoroughly validate their claims empirically.

---

> ### Author Response · Authors · 2021-11-19
> **Response to Reviewer HKa2**
>
> We would like to thank you for your feedback, and use this opportunity to respond to some of the comments and clarify any misunderstandings.
>
> > “I don’t agree with the dichotomy presented ... My understanding is “Neural ODE” refers to ODEs parameterized by neural networks.”
>
> Our original statement was to highlight the fact that in the past literature the term “neural ODE”  has sometimes been used to refer to a network which contains an ODE (with neural network system dynamics) rather than discrete layers. For example: [1] explicitly refers to these models as “neural ODEs”. Other times, “neural ODEs” refer to the actual ODE with neural network dynamics. To disambiguate, throughout our paper, we refrained from referring to the latter model as “neural ODEs”, but rather as “ODEs with dynamics parameterized by a neural network”. This is stated to clarify the terminology used, and does not alter the technical contributions of the work.
>
> We would like to thank you for these references. We have reviewed these papers and shall incorporate these
>
> >”Can you more thoroughly explain the benefits of asymptotically stable Neural ODEs?”
>
> Asymptotic stability ensures that trajectories from the system are similar when there are perturbations in initial condition, and eventually each trajectory will converge. For example, this is a property which is highly beneficial for generalization in robot imitation learning problems, where we have very few demonstrations, and we may wish to query trajectories starting from where no data has been observed. In these cases, enforcing stability will ensure that the integrated trajectory does not fly off. In particular, Fig 1 in [2] gives a demonstration of the benefits of learning a stable ODE system in generalizing robot motion. The requirement of being asymptotic stable is present in many robot learning set-ups including [3].
>
> > “Are there examples of Neural ODEs without diffeomorphisms that can be empirically demonstrated to be asymptotically unstable?”
>
> There are no asymptotic stability guarantees for the usual ODE parameterized by a neural network [4]. That is, if we integrate trajectories from two initial conditions which are close to each other, the trajectories may diverge.
>
> > “It doesn’t seem that this definition of asymptotic stability in Definition 4.3 permits periodic solutions, yet there are experiments with these dynamics. Can you explain this?”
>
> Our method provides us with a straightforward method of constraining the learned ODEs to be asymptotically stable, by constraining the eigenvalues of the base ODE. To clarify, our experiments were by default not constrained to be asymptotically stable, and only constrained in our robot motion generation experiments, where this is desired and we explicitly state that these are constrained to be asymptotically stable.
>
> > “A principal advantage ... is that solution curves can be obtained without evaluating the jacobian of the diffeomorphism. I’m not sure why this is a big advantage.”
>
> We would like to clarify a potential misunderstanding here. Solution curves can be evaluated directly by a forward and backward pass of the diffeomorphism: One pass to find the corresponding initial solution for the base ODE, using this we can integrate a trajectory of the base ODE, then another pass finds the trajectory of the target ODE.
>
> The advantage arises if we can use a more easy-to-integrate ODE as the base ODE. For example, if we restrict the base ODE to be linear, we can get trajectories in closed-form,  without using a numerical integrator. The passes of the diffeomorphism are relatively cheap relative to the sequential numerical integration, and can be batched. This gives us huge computational speed-up, as demonstrated in the experiments.
>
> > “Are there more fine-grained measures of time-cost ... Does Table 2 include results from models in Rubanova et al. 2019? Or is it just Linear ODE models integrated with different solvers?”
>
> Table 2 includes results from models in Rubanova et al., where the ODE model used in the experiments has dynamics models parameterised by a neural network, and a numerical integrator needs to be used. In these experiments, we use a linear ODE base, which is integrated in closed-form, and never numerically integrated with solvers. Thus, function evaluations would not quite be comparable, as integrating the linear base ODE does not require a numerical solver.
>
>
> [1] “ANODEV2: A Coupled Neural ODE Framework”, Zhang et al., 2019. https://papers.nips.cc/paper/2019/hash/227f6afd3b7f89b96c4bb91f95d50f6d-Abstract.html.
>
> [2] “Learning Contracting Vector Fields For Stable Imitation Learning”, Sindhwani et al, 2018. https://arxiv.org/pdf/1804.04878.pdf.
>
> [3] “Learning Partially Contracting Dynamical Systems from Demonstrations”, Ravichandar et al, 2017. http://proceedings.mlr.press/v78/ravichandar17a/ravichandar17a.pdf.
>
> [4] “Stable Neural Flows”, Massaroli et al, 2020. https://arxiv.org/pdf/2003.08063.pdf

---

> > ### Comment · Reviewer_HKa2 · 2021-11-21
> > **Response**
> >
> > > This is stated to clarify the terminology used, and does not alter the technical contributions of the work.
> > Thank you for clarifying this, I understand now. I think it might help to add some version of this explanation in the appendix.
> >
> > > There are no asymptotic stability guarantees for the usual ODE parameterized by a neural network [4].
> > I agree that there are no theoretical guarantees, but I'm curious if you have any experimental results showing that this asymptotic instability leads to issues such as worse generalization, or trajectories with different initial conditions diverging?
> >
> > Thanks for clarifying some of the other points of confusion I had.
> >
> > > It isn’t well-characterized when it is suitable to use a Linear Base ODE vs. a neural network ODE. When is the linear base ODE expressive enough? Are there tractable components that can be composed with sufficient expressivity, instead of having to resort to a neural network ODE?
> > I'm still curious about this component, and relatedly I think the experiments need to be more thorough to understand when the Linear Base ODE can be used to give a substantial speedup without hurting performance.

---

### Official Review · Reviewer_CkYu · 2021-11-02

**Correctness:** 4
**Technical Novelty And Significance:** 2
**Empirical Novelty And Significance:** 2
**Recommendation:** 6
**Confidence:** 3

**Main Review:**

The strengths of the paper are:
- it follows a series of work in this direction of ODE estimation, which has been validated as an important direction for ML
- it is mathematically sound
- well written

The weaknesses are:
- from the limited experiments presented it is not clear how applicable the method is for complex practical problems
- existence of an diffeomorphism is a critical aspect of the paper, it is not clear if such a tractable mapping that can be estimated using using a NN can really exist
- related to the above, the choice of the base ODE is another aspect that might be non-trivial

**Summary Of The Paper:**

The paper has been well summarized by the authors in their abstract. The following is taken directly from there:

Learning ordinary differential equations (ODEs), where a flexible function approximator (often a neural network) is used to estimate the system dynamics, given as a time derivative has become popular after the publication of the paper by Chen et. al. at NeurIPS 2018.. However, these integrators can be unsatisfactorily slow and unstable when learning systems of ODEs from long sequences. In this work, the authors propose to learn an ODE of interest from data by viewing its dynamics as a vector field related to another base vector field via a diffeomorphism. By learning both the diffeomorphism and the dynamics of the base ODE, the authors provide an approach to offload some of the complexity in modelling the dynamics directly on to learning the diffeomorphism. Consequently, by restricting the base ODE to be amenable to integration, we can speed up and improve the robustness of integrating trajectories from the learned system.

**Summary Of The Review:**

Clearly, the paper adds to the existing literature and is interesting. It is well written and mathematically sound.  It is however, unclear how general the approach is and that brings into question the significance of the contribution.

---

> ### Author Response · Authors · 2021-11-19
> **Response to Reviewer CkYu**
>
> We would like to thank you for your positive feedback, and use this opportunity to respond to some of the comments and clarify any misunderstandings.
>
> > “from the limited experiments presented it is not clear how applicable the method is for complex practical problems”
>
> We'd like to clarify to the reviewer that, in our experiments, we have used our approach to learn ODEs both within a neural ODE framework on machine learning problems (Table 2), on a variety of baseline systems (Lotka-Volterra, Lorentz, Robertson), and in particular, in robots problems on real world robotics data (see "A Time Critical Application—Robot Motion Generation as Stable ODE Learning" in section 5.1).
>
> > “existence of an diffeomorphism is a critical aspect of the paper, it is not clear if such a tractable mapping that can be estimated using using a NN can really exist”
>
> We refer to equations 4 and 5 in section 4.2 which outline how the bijections can be learned. In other words, the NN is set up such as it is a diffeomorphism _by definition_. We would like to emphasise that models for learning differentiable bijection have been well-studied in the normalizing flow for density estimation literature, these include models outlined in [1] and [2].
>
> [1] Laurent Dinh, Jascha Sohl-Dickstein, and Samy Bengio. Density estimation using real NVP. In International Conference on Learning Representations (ICLR), 2017.
>
> [2] Lynton Ardizzone, Jakob Kruse, Sebastian Wirkert, Daniel Rahner, Eric W. Pellegrini, Ralf S. Klessen, Lena Maier-Hein, Carsten Rother, and Ullrich Köthe. Analyzing inverse problems with invertible neural networks. In International Conference on Learning Representations (ICLR), 2019.

---

### Decision · Program_Chairs · 2022-01-20

**Decision:**

Reject

**Comment:**

This submission proposes a new manner to learn ordinary differential equations, aiming to improve their efficiency. While judging it interesting, the reviewers are quite split on this work. Overall there was no strong consensus to accept, nor anyone willing to champion this work.

The main stated weaknesses are

- The reliance on the existence of a diffeomorphism (and its choice in the method)
- The choice of the base and its expressiveness
- A somewhat limited experimental section, not indicating strongly how amenable this would be to more complex problems.